# Titanium Dioxide Nanotubes as Solid-Phase Extraction Adsorbent for the Determination of Copper in Natural Water Samples

**DOI:** 10.3390/ma15030822

**Published:** 2022-01-21

**Authors:** Bochra Bejaoui Kefi, Imen Bouchmila, Patrick Martin, Naceur M’Hamdi

**Affiliations:** 1Laboratory of Useful Materials, National Institute of Research and Pysico-Chemical Analysis (INRAP), Technopark of Sidi Thabet, Ariana 2020, Tunisia; bouchmila.imen.ic@gmail.com; 2Department of Chemistry, Faculty of Sciences of Bizerte, University of Carthage, Zarzouna 7021, Tunisia; 3Transformations & Agroressources Unity, UR7519, Université d’Artois-UniLaSalle, F-62408 Bethune, France; 4Research Laboratory of Ecosystems & Aquatic Resources, National Agronomic Institute of Tunisia, Carthage University, Tunis 1082, Tunisia; naceur_mhamdi@yahoo.fr

**Keywords:** atomic absorption spectroscopy, copper, hydrothermal treatment, solid phase extraction, TiO_2_ nanotubes

## Abstract

To increase the sensitivity of the analysis method of good copper sample preparation is essential. In this context, an analytical method was developed for sensitive determination of Cu (II) in environmental water samples by using TiO_2_ nanotubes as a solid-phase extraction absorbent (SPE). Factors affecting the extraction efficiency including the type, volume, concentration, and flow rate of the elution solvent, the mass of the adsorbent, and the volume, pH, and flow rate of the sample were evaluated and optimized. TiO_2_ nanotubes exhibited their good enrichment capacity for Cu (II) (~98%). Under optimal conditions, the method of the analysis showed good linearity in the range of 0–22 mg L^−1^ (R^2^ > 0.99), satisfactory repeatability (relative standard deviation: RSD was 3.16, *n* = 5), and a detection limit of about 32.5 ng mL^−1^. The proposed method was applied to real water samples, and the achieved recoveries were above 95%, showing minimal matrix effect and the robustness of the optimized SPE method.

## 1. Introduction

Inorganic contaminants are generally found in trace and ultra-trace levels to a complex matrix. To analyze them, preliminary treatment of samples containing very complex matrix is thus necessary. Indeed, this step is crucial in the analytical process since it affects the quality of the analysis in terms of sensitivity, precision, and repeatability. Several pretreatment methods have been developed, such as liquid–liquid extraction, supercritical phase extraction, microwave extraction, and solid-phase extraction (SPE) [1,2,3,4,5,6,7,8,9].

Recently, many researchers preferred the SPE method because of its advantages to the enrichment, high recovery, simplicity, rapid, and low organic solvent consumption. SPE is selective, efficient, clean, and can be automated or even used online with several analysis techniques [10,11,12,13,14,15,16]. This explains the constant development of this technique through the search for new adsorbents. The most used adsorbents, such as commercial activated carbon and ion-exchange resins [17,18,19,20], are characterized by their high production and regeneration costs, which has developed the research toward other adsorbents such as metal-organic frameworks (MOFs), nanocomposites and nanomaterials [10,11,12,21,22,23,24,25,26,27,28,29,30,31]. The nanometric size of these structures leads to an increase in the proportion of atoms present on their surface, thus allowing a high reactivity and an interesting adsorption capacity [32,33]. The most used of these nanomaterials are carbon nanotubes [10,11,12,22,24,34,35]. Nevertheless, nanostructured titanium dioxide proves to be a possible alternative to the rather high cost of these nanomaterials since its elaboration is easy to implement and inexpensive. Several morphologies of nanomaterials have been tested, among them the nanotubular one which has an open mesoporous morphology that gives them the properties of adsorbents [36,37]. The typical value of the total volume of specific pores of nanotubes is in the range of 0.60 to 0.85 cm^3^ g^−1^. They have a large specific surface area compared to the starting materials (~50 m^2^ g^−1^) and whose value determined by Brunauer-Emmett-Teller [BET] method varies from 100 to 400 m^2^ g^−1^ [38,39,40,41,42]. The values of the total volume and specific surface area depend on the average diameter of the nanotubes. Among these nanotubular structures, titanium dioxide nanotubes can be used for the same applications as those known for the precursor TiO_2_ before the transformation. Their new morphology is also accompanied by new properties allowing particular and specific applications in catalysis [41,42,43,44], medical applications [45,46], environmental applications, and extraction processes of various organic and inorganic contaminants: pesticides, polycyclic aromatic hydrocarbons (PAHs), phthalates, dyes, and heavy metals [47,48,49,50]. 

Among the inorganic contaminants, we are interested in the element copper. Copper is a multipurpose metal, when present in low doses, it is a trace element essential to life, but in higher doses, it is toxic [51,52,53,54,55]. As it is not biodegradable, it can accumulate and eventually reach high levels [53,55]. Its origin in water is very diverse, in addition to the natural contents coming mainly from the deposits, its use by man causes important discharges in the environment [55]. Research in recent years has been based on the identification of new processes of analysis and even good methods for removing copper from water [22,26,27,34].

For Cu^2+^ analysis, atomic absorption spectroscopy (AAS) and inductively coupled plasma are the most widely used [4]. Other more sensitive and selective methods are also used. Qi et al. applied sensor-based methods for rapid detection of Cu^2+^ in water [56].

Therefore, in this work, a solid phase adsorbent, titanium dioxide nanotubes, was synthesized to develop a solid phase extraction (SPE) method for preconcentration/extraction of Cu (II) from natural water.

## 2. Materials and Methods

### 2.1. Experimental 

Chemical Reagents and Solutions

The TiO_2_ nanotubes were elaborated using a precursor of titanium dioxide (P25, Degussa-Hüls A.G. (Evonik, Germany), 68% anatase and 32% rutile, SBET = 50 m^2^ g^−1^, non-porous and pHPZC = 5.6), a concentrated solution of caustic soda (Fisher Chemicals, Ohio, USA, Purity: 98.64%) and a diluted solution of hydrochloric acid (Panreac Quimica S.A., Barcelona, Spain, Purity: 37%). The nitric acid was purchased from Scharlau Chemie S.A., (Barcelona, Spain, purity 65%) and the water used throughout the elaboration is ultra-pure purchased from Milli-Q, Millipore (Merck Millipore, MA, USA), 18 MΩ cm. The hydrated copper nitrate Cu(NO_3_)2 × 3H_2_O was purchased from the company Panreac Quimica S.A. (Barcelona, Spain) with 99% purity. The different Cu (II) solutions are prepared by dissolving adequate amounts of hydrated copper nitrate in ultra-pure water.

### 2.2. Apparatus

The copper was determined by a flame atomic absorption spectrometer model Analytik Jena “novAA 400”. (Jena, Germany).The pressure and flow rate of the nebulizer were 3 bars and 0.45 L min^−1^, respectively. The pump speed was 22 rpm and the generator power was maintained at 40.68 MHz. The studied element was identified at the wavelength of 324.754 nm.

#### 2.2.1. Synthesis of Titanium Dioxide Nanotubes

The alkaline hydrothermal method, as described in a previous study [38,39,49] was used to prepare the titanate nanotubes. A commercial TiO_2_ (P25, 0.50 g) was dispersed in a 15 mL of NaOH solution of 11.25 mol L^−1^ and introduced into a Teflon-lined autoclave with an 80% filling factor. The autoclave was then heated at 130 °C for 20 h to prepare the hydrogenated nanotubes (HNT) samples. This later was washed with one liter of hot ultra-pure water then filtered under vacuum to eliminate the excess of unreacted soda. To exchange the Na^+^ ions contained in the structure by protons, we proceeded to the neutralization of the product by a solution of hydrochloric acid (0.1 mol L^−1^). The precipitate was then washed with 0.5 L of hot ultra-pure water to remove NaCl formed in excess. The obtained wet solids were dried in an oven at 80 °C for 24 hr. Finally, HNT was calcined at 500 °C for two hours in an air atmosphere to obtain titanium dioxide nanotubes (TON). The latter were used as adsorbents for Cu (II) removal from water samples.

#### 2.2.2. Characterization of Nanotubes

The structural study of HNT and TON was determined at room temperature with X-ray diffraction (XRD) analyses using an automated “X’Pert PRO MPD, PANalytical Co., Almelo, The Netherlands” diffractometer X-ray diffraction (XRD). Monochromatic Cu Kα-radiation (λ = 1.5418 Å) was obtained with a Ni-filtration and a system of diverging and receiving slides of 0.5° and 0.1 mm, respectively. The diffraction pattern was measured with a voltage of 40 kV and a current of 30 mA over a 2θ range of 3–40° using a step size of 0.02° at a scan speed of 1 s per step.

For a porous solid, the term texture mainly refers to the specific surface area (SBET, m^2^ g^−1^), specific pore volume (V_p_, cm^3^ g^−1^), porosity ε, pore shape, and pore distribution or distribution of pore volumes as a function of pore size. The specific surface areas of HNT and TON, pore volume, and average pore diameters were determined by the BET method and calculated from the nitrogen adsorption-desorption measurements at 77 K using the “Micro metrics ASAP 2000” volumetric apparatus. 

Morphology and structure of titanium dioxide nanotubes were characterized by transmission electron microscopy (TEM), purchased from Philadelphia (PA, USA) with an acceleration current of about 200 kV and by high-resolution transmission electron microscopy (HR-TEM) of the “JEOL-2010” type (the acceleration current is 400 kV)

#### 2.2.3. pH of Points Zero Charges (pHpzc)

The pH of zero charge point (pHpzc) of the studied material is an important parameter in adsorption phenomena that depends on electrostatic forces. It gives information on the charge of dominant sites on the solid surface. Indeed, the surface is positively charged at pH values below pHpzc and negatively charged at pH values above pHpzc. To measure the pHpzc values, the salt method was applied [57]. The titration cell is filled with 100 mL of electrolyte NaOH (0.01 mol L^−1^)/NaCl (0.1 mol L^−1^) and 1 g of the studied material (HNT and TON). The material suspension was equilibrated for 24 h, and then the titration was carried out with 0.1 mol L^−1^ of HCl while measuring the pH for each volume added. Blank titration is done without studied material in the same way as in the presence of the solid with the same concentration of electrolytes.

#### 2.2.4. SPE Process 

Empty solid-phase extraction cartridges (0.2 g, 3 mL, polypropylene) are filled with TON. This adsorbent is held in place by the polypropylene upper and lower frits. The packed cartridge was then placed on a vacuum elution apparatus. This solid phase was first conditioned with 10 mL HNO_3_ (2 M) and 10 mL of ultra-pure water. After that, a volume of ultra-pure water or sample water varying from 100 to 500 mL and spiked with Cu (II) at a concentration of 2.62 mg L^−1^ is percolated through the cartridge at a flow rate to be determined during this work. The washing of the adsorbent, with 10 mL of ultra-pure water is performed only in the case of real water samples. Finally, to ensure complete desorption of Cu (II) three elution solvents of volume 10 mL were tested namely nitric acid, hydrochloric acid, and pure ethanol.

Factors influencing this method (the elution solvent, volume, and concentration, the volume of the sample, the mass of the adsorbent, the pH of the sample, elution, and percolation flow rates) were thus optimized. The extraction yield of Cu (II) cations is calculated according to the following Equation (1):(1)$$R\%=\frac{C_{f}-C_{i}}{C_{0}}\cdot 100$$
with:

Cf = concentration of Cu (II) obtained from the spiked sample;

Ci = concentration of Cu (II) obtained from the unspiked sample;

C_0_ = concentration of the Cu (II) added to the sample: spiking level: 2.62 mg L^−1^.

## 3. Results and Discussion 

### 3.1. Material Characterizations 

The change in the crystal structure from HNT to calcined TON was studied by XRD (Figure 1). The comparison of the HNT XRD pattern with the ASTM sheet corresponding to TiO_2_ (P25), showed that the elaborated nanotubes did not correspond to either anatase or rutile. HNT showed an orthorhombic system with the lattice constants a_0_ = 1.926 nm, b_0_ = 0.378 nm, and c_0_ = 0.300 nm [39,58] and according to the ASTM sheet N° 47–0124, its structure corresponded well to that of hydrogenated nanotubes of type H_2_Ti_2_O_5_ × H_2_O. Upon calcination of HNT at 500 °C, the XRD pattern of TON showed characteristic peaks of anatase phase TiO_2_ and additional peaks, which corresponded to H_1_._2_Na_0.8_O_7_Ti_3_ crystallizing in the monoclinic system. 

The specific surface area (S_BET_), pore volume (V_p_), as well as an average diameter (d_p_) of nanotubes before and after calcination are summarized in Table 1. When TiO_2_ particles (P25) are transformed into hydrogenated nanotubes, a considerable increase in the specific surface area from 50 to 269 m^2^ g^−1^ was observed. Indeed, the multi-walled structure of the elaborated nanotubes HNT gives them a high specific surface since nitrogen molecules can intercalate in the interfoliar spaces. When calcined, the specific surface of nanotubes decreased, and the pore diameter increased (from 9 nm for T = 130 °C to 23 nm for T = 500 °C). The decrease in S_BET_ after calcination can be explained by: (i) The disappearance of the multi-walled structure of the nanotubes after calcination, thus the measured S_BET_ corresponds only to the inner and outer surface of the nanotubes; (ii) the decrease of the number of nanotubes after calcination.

The analysis of the TEM and HR-TEM images of the HNT nanotubes (Figure 2), shows their hollow and homogeneous nanotubular structure. The outer diameters of tubes were 6–8 nm and inner diameters were 4–6 nm, and the lengths were measured to be several hundred nanometers. The HR-TEM analysis showed that the walls of the nanotubes were amorphous (Figure 2b).

After calcination, the outer and inner diameters of nanotubes decreased to 7–5 nm (Figure 3a). However, HR-TEM analysis was consistent with the XRD results and showed nanotubes walls transformed into crystalline anatase (Figure 3b).

The anatase TiO_2_ nanotubes (TON) were used as adsorbents for the SPE of Cu (II) from water samples. Factors that may influence the extraction yields of the copper ions namely the type, volume, concentration, and flow rate of elution solvents, were studied. Likewise, the volume, pH, and flow rate of percolated samples and the effect of the mass of the adsorbent were optimized. 

### 3.2. Optimization of SPE Method

#### 3.2.1. Effect of Elution Solvent Type and Volume

The influences of three eluents (nitric acid, hydrochloric acid, and ethanol) on the recoveries of Cu ions from TiO_2_ nanotubes were examined (Figure 4). 

They are the most used solvents for solid-phase extraction of heavy metals [12,25,34,47]. As can be seen in Figure 4, the extraction yield of Cu cations is strongly related to the nature of the elution solvent. The maximum elution is obtained with HNO_3_ (2M) [34], therefore it was chosen as elution solvent for the next steps of this optimization. This result is in agreement with that presented by Soylak et al. [34], who studied the extraction of Cu (II) by carbon nanotubes. 

The effect of the volume of HNO_3_ on the recovery of the Cu (II) was investigated in the range 2–14 mL maintaining its concentration at 2M. The results are shown in Figure 5. 

As can be seen, the volume of HNO_3_ had a significant effect (*p* < 0.001) on the efficiency of the SPE method, and on the extraction yield of Cu (II) cations. Extraction yield increases progressively with the volume of HNO_3_ until reaching a maximum of 10 mL. To ensure a maximum elution of Cu (II) cations, a volume of 12 mL of HNO_3_ was used for the next steps of the optimization.

#### 3.2.2. Effect of Eluent Concentration and Flow Rate

Several factors affecting the elution efficiency of analytes, such as eluent concentration, and flow rate were also studied. HNO_3_ concentrations between 0.5 and 4 mol L^−1^ were examined, and their effects on the extraction efficiency of Cu (II) cations are shown in Figure 6. 

Obtained results showed insignificant differences in recoveries among the different concentrations of HNO_3_. Therefore, the acidic solvent concentration of 2 mol L^−1^ was chosen for the extraction of Cu cations.

Considering the importance of the elution or desorption step in the SPE process, several elution flow rates from 1 to 8 mL min^−1^ were also tested (Figure 7). 

Results show a decrease of the extraction yields when the elution flow rate exceeds 2 mL min^−^^1^.

#### 3.2.3. Effect of Sample Volume and Flow Rate

The breakthrough volume in the solid phase extraction was investigated. Different volumes from 100 to 500 mL of ultra-pure water spiked with 2.62 mg L^−1^ of Cu (II) was percolated through the cartridge. Results presented in Figure 8 show a decrease in yields as the percolated volume increased. 

Therefore, a sample volume of 100 mL was recommended to obtain the maximum extraction yield. 

To ensure better repeatability of results, the flow rate of the sample percolation must be controlled. For this purpose, percolation flow rates ranging from 1 to 8 mL min^−1^ were tested, with the other conditions kept constant. Results are given in Figure 9 and they illustrate maximum extraction yield when the sample was percolated at a flow rate lower than 2 mL min^−1^.

#### 3.2.4. Effect of Sample pH

The pH is an important factor in the SPE procedure [57,59]. It determines the surface charge of the TiO_2_ nanotubes, on the one hand, and on the other hand the form of copper in solution. Indeed, the pH values of precipitation of Cu (II) hydroxide, correspond to initial concentrations between 2.62 and 1500 mg L^−1^ and varies from 6.89 to 5.80 respectively. The values of pH at the point of zero charge pHpzc are frequently used to determine the sorption properties of oxides and hydroxides [60]. Results from titration curves (Figure 10) showed pHpzc of 6.7 and 8.3 for HNT and TON, respectively. 

The pHpzc of the used adsorbent TON is 8.3, this value shows that the surface is positive for a pH value lower than 8.3 and negative for pH above 8.3.

The effect of the sample pH on recoveries of Cu (II) was examined between 3 and 9. The results presented in Figure 11 show lower adsorption capacities at pH values below 4, the is possibly due to the presence of excess H^+^ ions competing with Cu (II) ions for the available adsorption sites. Then this adsorption capacity reached its maximum values when the pH was between 5 and 6 [49,59].

So, we can conclude that the adsorption of copper was more pronounced in the case of positive charges of TON adsorbent. 

#### 3.2.5. Effect of the Mass of the Adsorbent

The mass of the adsorbent, which is a determining factor in the solid phase extraction method, is also optimized. TON amounts between 0.1 and 0.5 g were tested for the extraction of copper cations. The results given in Figure 12 show insignificant differences between calculated yields.

So, with a minimum mass of nanotubes, 0.1 g, maximum recovery of ~100 % can be achieved. It is the nanometric size of these structures and the large proportion of atoms present on their surface that favor this high reactivity.

### 3.3. Optimal Conditions

The SPE procedure following the optimal conditions obtained is therefore summarized as follows: through a cartridge, filled with 0.1 g of TiO_2_ nanotubes, 10 mL of HNO_3_ (2 mol L^−1^), and 10 mL of ultrapure water are first percolated. A sample volume of 100 mL of water is percolated through the adsorbent at a flow rate of 2 mL min^−1^. The washing of the adsorbent was performed with 10 mL of ultrapure water (in the case of real water samples). Elution of Cu (II) is performed with 12 mL HNO_3_ (2 M) at a flow rate of 2 mL min^−1^. These optimum conditions allow an extraction yield higher than 95% and a maximum adsorption capacity of 70 mg g^−1^ of TON. This adsorption capacity is perfect when TiO_2_ nanotubes are applied as SPE adsorbent. While, for environmental applications and water treatment, higher adsorption capacities are desired [30,61].

### 3.4. Characteristics of the Method

The analytical parameters such as the sensitivity, detection limits, and reproducibility of the analytical method can be improved with a good sample extraction method. Indeed, it allows a better sensitivity of the technique of analysis by removing interferents from the matrix. The extraction of Cu (II) cations with TiO_2_ nanotubes is repeated five times according to the optimal conditions obtained. Extraction yields were between 95.81 and 98.85% with satisfactory RSD of 3.16% (lower than 5%). Detection limit of AAS technique was in the range of 32.5 ng mL^−1^.

In the present work, flame atomic absorption spectrometry was used for the identification of Cu (II) cations. To consider all concentrations of the analyzed copper during the optimization of the SPE method, three calibration ranges are considered (Table 2). The study of linearity showed a linear relationship between the concentration and the response of the technique. Calibration equations and correlation coefficients are illustrated in Table 2 and results show good and linear correlation of the regression method R^2^ > 0.998.

### 3.5. Application of Optimal Conditions to Real Water Samples

To evaluate the effect of the matrix on the extraction efficiency of the optimized SPE method, the optimal conditions were applied to real water samples (tap water and mineral water). Mineral water was spiked with Cu (II) at a concentration of 2.37 mg L^−1^, while tap water was spiked at four Cu (II) concentration levels. Analytical results given in Table 3 show the minimal effect of the matrix.

Calculated yields were between 87 and 97% and were not far from those determined with ultra-pure water.

## 4. Conclusions

In this study, a reliable and efficient method was used for the determination of Cu (II) in water samples. This method consists of using well-characterized titanium dioxide nanotubes TON as an adsorbent in the solid phase extraction procedure of this analyte. TON showed interesting results in terms of calculated extraction yields that are higher than 95% and repeatability since the calculated RSD value is about 3.16%. Furthermore, this method improved the detection limit of the analysis technique of copper (32.5 ng mL^−1^). The optimal conditions determined during this work were successfully applied for the solid-phase extraction of Cu (II) from real water samples and a minimal matrix effect is observed. The simplicity and relative affordability of the preparation of titanium dioxide nanotubes, as well as their effectiveness as SPE adsorbents, give them great potential in the fields of selective inorganic contaminant separation and sample process.

## Figures and Tables

**Figure 1 materials-15-00822-f001:**
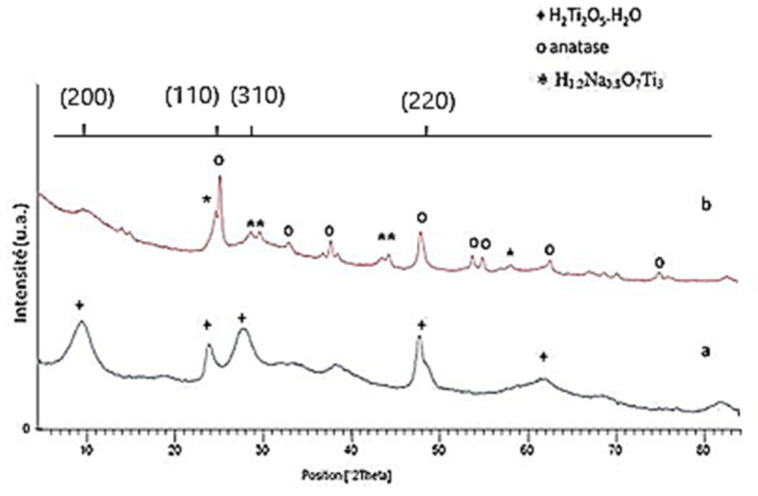
XRD of orthorhombic H_2_Ti_2_O_5_ phase (**a**) and calcined TiO_2_ nanotubes (TON) (**b**).

**Figure 2 materials-15-00822-f002:**
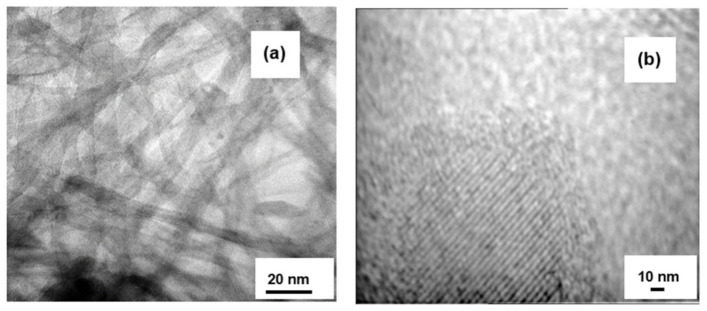
Images of HNT nanotubes with (**a**) TEM and (**b**) HR-TEM.

**Figure 3 materials-15-00822-f003:**
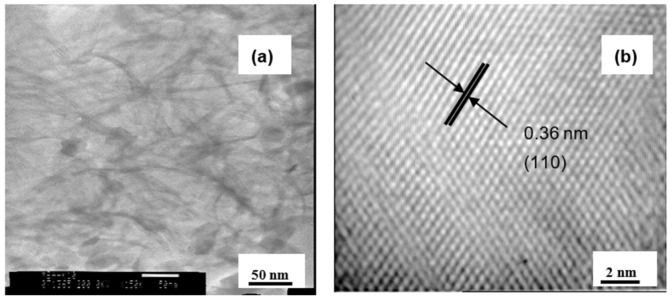
Images of TON nanotubes with (**a**) TEM and (**b**) HR-TEM.

**Figure 4 materials-15-00822-f004:**
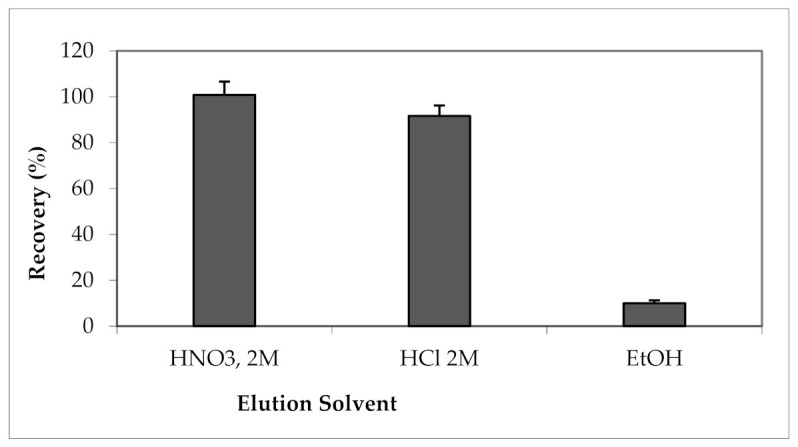
Effect of the elution solvent type on the extraction yield of Cu (II). Solvent volume = 10 mL.

**Figure 5 materials-15-00822-f005:**
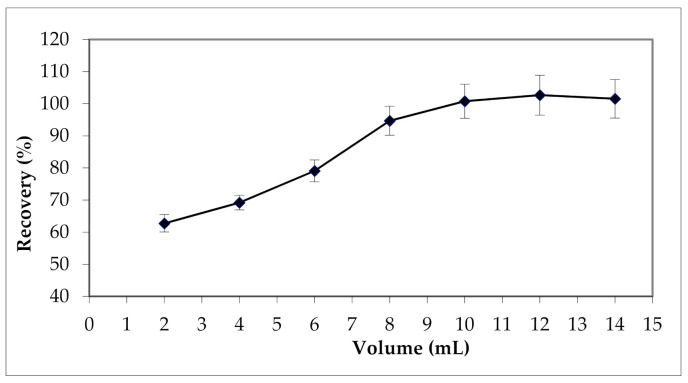
Effect of HNO_3_volume on Cu (II) extraction.

**Figure 6 materials-15-00822-f006:**
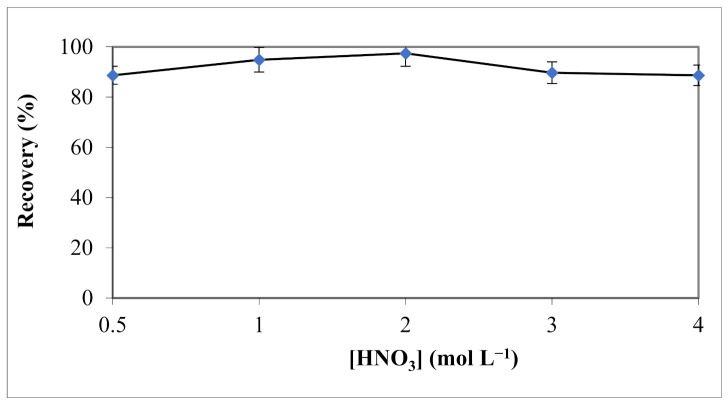
Effect of HNO_3_ concentration on Cu (II) extraction (solvent volume = 12 mL).

**Figure 7 materials-15-00822-f007:**
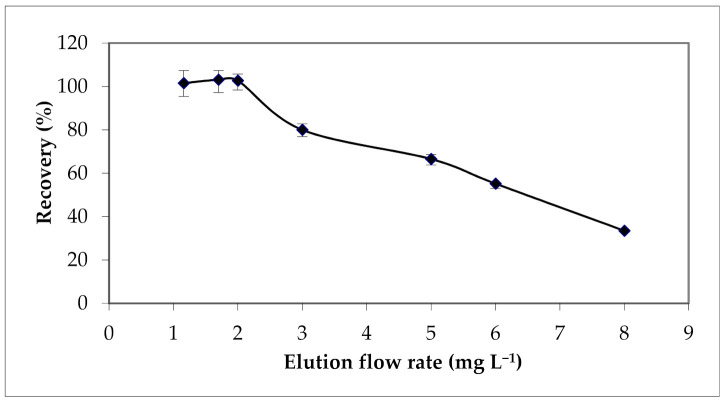
Effect of elution flow rate on the extraction yield of Cu (II). Solvent volume = 12 mL; [HNO_3_] = 2 M.

**Figure 8 materials-15-00822-f008:**
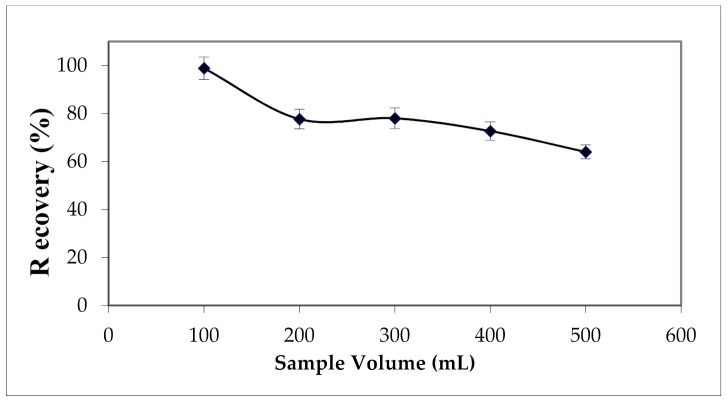
Effect of sample volume on the extraction yield of Cu (II). Solvent volume = 12 mL; [HNO_3_] = 2 M; the elution flow rate = 2 mL min^−1^.

**Figure 9 materials-15-00822-f009:**
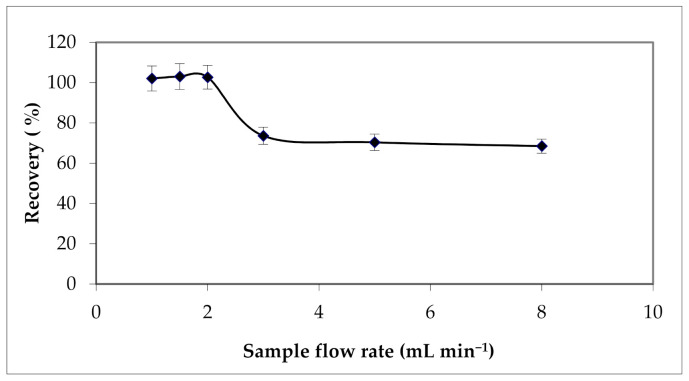
Effect of percolation flow rate on Cu (II) extraction yield. Solvent volume = 12 mL; [HNO_3_] = 2 mol L^−1^; the elution flow rate = 2 mL min^−1^; sample volume = 100 mL.

**Figure 10 materials-15-00822-f010:**
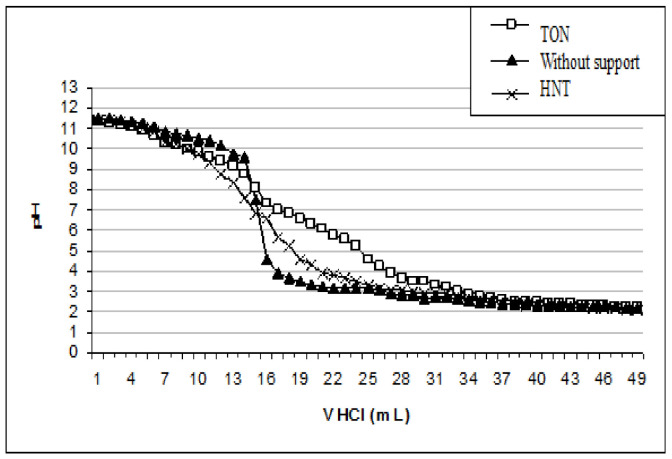
Titration curves of NaOH (0.01 mol L^−1^)/NaCl (0.1 mol L^−1^) solution by HCl (0.1 mol L^−1^) in the absence and presence of 1 g of the material (HNT et TON).

**Figure 11 materials-15-00822-f011:**
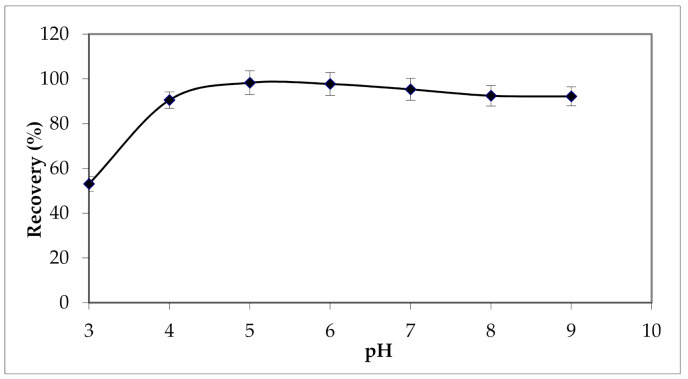
Effect of the sample pH on solid-phase extraction of Cu (II). Solvent volume = 12 mL; [HNO_3_] = 2 M; the elution flow rate = 2 mL min^−1^; sample volume = 100 mL; percolation flow rate = 2 mL min^−1^.

**Figure 12 materials-15-00822-f012:**
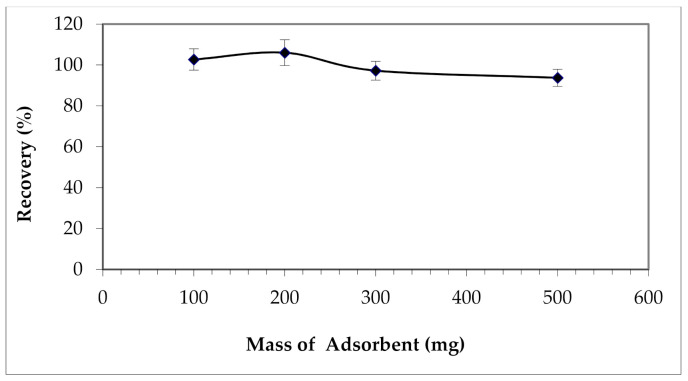
Effect of the mass of the adsorbent on Cu (II) extraction.

**Table 1 materials-15-00822-t001:** Textural characteristics of elaborated nanotubes before and after calcination.

T (°C)	S_BET_ (m^2^ g^−1^)	V_p_ (cm^3^ g^−1^)	d_p_ (nm)
130	269	0.67	9
500	~100	0.63	23

S_BET_: specific surface, V_p_: porous volume, d_p_: average pore diameter.

**Table 2 materials-15-00822-t002:** Calibration equation, correlation coefficient of Cu (II) analyses by SAAF.

Calibrations Range (mg L^−1^)	Calibration Equation	R^2^
**0–1.65**	Y = 0.0869x − 0.0017	0.9982
**0–3**	Y = 0.0628x − 0.0022	0.9999
**0–22**	Y = 0.0131x − 0.0015	0.9985

**Table 3 materials-15-00822-t003:** Recovery percentages of tap and mineral water samples spiked with Cu cations.

Water Samples	Added Copper (mg L^−1^)	R%
Tap water	2.37	97.04
	1.57	95.23
	0.68	89.54
	0.26	87.21
Mineral water	2.37	96.20

## Data Availability

All the data is available within the manuscript.

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
