# Peer review of "Titanium Dioxide Nanotubes as Solid-Phase Extraction Adsorbent for the Determination of Copper in Natural Water Samples"

_materials, 2022, doi:10.3390/ma15030822_

Round 1
Reviewer 1 Report
The authors have made an attempt to apply Titatnium Dioxide nanotubes for determination of copper in water which is some what good to contribute for water management and development.
The authors can present the data with more technical clarity.
The authors have to give a comparative data how Titatnium Dioxide nanotubes show superior performance than any other nanoparticles/composites?
Author Response
Point1: The authors have made an attempt to apply Titatnium Dioxide nanotubes for determination of copper in water which is somewhat good to contribute for water management and development.
Response 1:
The large increases in population, water scarcity problems and dramatic climate change patterns have already caused flooding and drought, have made drinking water a competitive resource in many parts of the world.
Water management and wastewater treatment is a major challenge in recent research. In this context, several new methods and materials are introduced to address the issue of water purification.
The objective of the present work is the development of a new method of treatment of water samples for analytical purposes in conjunction with the water quality control.
For wastewater treatment, nanomaterials in general, remain expensive materials and their elaboration cost prevents and limits their large-scale use. On the other hand, they can be used as a filter cartridge due to their high adsorption rate of several heavy metals.
Point2: The authors can present the data with more technical clarity.
Response 2:
More technical information and more interpretation are added in the text
Point3: The authors have to give a comparative data how Titatnium Dioxide nanotubes show superior performance than any other nanoparticles/composites?
Response 3:
Comments and comparative data are added to the section 3.3

Reviewer 2 Report
This manuscript presented the solid-phase extraction sampling of Cu2+ ions using TiO2 nanotubes as adsorbent. The influence of concentration, pH value, flux rate are characterized. A detection limit of 32.5 ngml-1 was reported. These results are attractive. However, the novelty and detailed procedures should be improved. The following items should be elucidated.
- In the introduction, the organic absorbance of Cu2+, such as L- -cysteine (Qi et al. Electrophoresis 40 (2019) 2699-2705) should be reviewed and compared.
- Explain why the recovery decreases in an acidic solution.
- Data of Table 2 for fitting should be displayed in the figure
- How to determine the detection limit of 32.5 ng.ml-1?
- How about the absorption capacity of TiO2 NTs of Cu2+? How about the comparison to other nano oxides (such as Fe2O3, see Wu et al. Chemosphere 230 (2019) 527)?
- What are the limit factors of absorption capacity of TiO2 NTs? How to further improve it?
Author Response
Point1: In the introduction, the organic absorbance of Cu2+, such as L- -cysteine (Qi et al. Electrophoresis 40 (2019) 2699-2705) should be reviewed and compared.
Response 1:
This paragraph is added in the introduction:
For Cu2+ analysis, atomic absorption spectroscopy (AAS) and Inductively Coupled Plasma are the most widely used [4]. Other more sensitive and selective methods are also used. Qi et al. applied sensor-based methods for rapid detection of Cu2+ in water [Qi, H., Zhao, M., Liang, H., Wu, J., Huang, Z., Hu, A., ... & Zhang, J. (2019). Rapid detection of trace Cu2+ using an l‐cysteine based interdigitated electrode sensor integrated with AC electrokinetic enrichment. Electrophoresis, 40(20), 2699-2705]
Point2: Explain why the recovery decreases in an acidic solution.
Response 2:
The adsorption capacity decreases at pH values below 4, this is possibly due to the presence of excess H+ ions competing with Cu(II) ions for the available adsorption sites.
More interpretation is added to this section
Point3: Data of Table 2 for fitting should be displayed in the figure
Response 3:
the calibration curves and their parameters (calibration equation and R2) are determined from the AAS technique software
Point4: How to determine the detection limit of 32.5 ng.ml-1?
Response 4:
The LDD was determined from regression lines determined for Cu(II) (calibration range 0 – 1.65 mg L-1 repeated 3 times) and Y-intercepts were considered as blank responses.
Point5: How about the absorption capacity of TiO2 NTs of Cu2+? How about the comparison to other nano oxides (such as Fe2O3, see Wu et al. Chemosphere 230 (2019) 527)?
Response 5:
Adsorption capacity is added in the section 3.3
And Fe-based ceramic nanocomposite membranes are added as novel adsorbent in the introduction
Point6: What are the limit factors of absorption capacity of TiO2 NTs? How to further improve it?
Response 6:
Adsorption capacity of TiO2 nanotubes depends on its mass, the concentration of copper and medium pH
Adsorption capacity of 70 mg g-1 of TON was attaint
this can be improved by enlarging the surface area, controlling the pore size and surface functionalization.

Reviewer 3 Report
This manuscript demonstrates the synthesis of titanium dioxide nanotubes as solid-phase extraction adsorbent for copper. However, it focused on the adsorption and recovery of Cu(II) rather than determination because this manuscript used a flame atomic absorption spectrometer for copper determination. Therefore, the manuscript really needs a major revision before publication. Specific comments can be found as below:
- Typo and grammatic errors in the Introduction should be carefully revised. For example, Line 36, Line 37, Line 52-54.
- The idea of this manuscript is to develop an adsorbent for Cu(II) capture rather than a new method to determine it. So, the introduction should be revised to reflect this topic. The abstract, discussion, and conclusion should be also reorganized.
- Figure 6, unit is wrong.
- The adsorption capacity of Cu(II) over TON should be examined and discussed. Typical procedure and analysis models can be found in the following paper: https://doi.org/10.1007/s12274-021-3918-6.
- Comparison with conventional solid phase should be added and discussed.
Author Response
Point1: Type and grammatic errors in the Introduction should be carefully revised. For example, Line 36, Line 37, Line 52-54.
Response 1:
Modification are made in the text
Point2: The idea of this manuscript is to develop an adsorbent for Cu(II) capture rather than a new method to determine it. So, the introduction should be revised to reflect this topic. The abstract, discussion, and conclusion should be also reorganized.
Response 2:
The aim of this work is to test the efficiency of titanium oxide nanotubes as a SPE adsorbent of copper from water samples. The copper is adsorbed and retained on this adsorbent (percolation step) and then desorbed and eluted (elution step). So, from sample complex matrix, an extract is obtained where copper is predominant with minimum of interferents and thus can improve the detection of this element by the analysis technique. On the other hand, these experimental conditions ensure the recovery of copper in small volumes of extraction solvent, which further increases the concentration of this element while remaining it stable and in free form. This application could be applied in the context of water quality control at the laboratory level.
Point3: Figure 6, unit is wrong.
Response 3:
Point4: The adsorption capacity of Cu(II) over TON should be examined and discussed. Typical procedure and analysis models can be found in the following paper: https://doi.org/10.1007/s12274-021-3918-6.
Response 4:
Metal-organic frameworks are cited as copper adsorbent in the introduction
And adsorption capacity is discussed in section 3.3
Point5: Comparison with conventional solid phase should be added and discussed.
Response 5:
Adsorption capacity is discussed in section 3.3

Reviewer 4 Report
The present work by Kefi et al. considers the synthesis of titania nanotubes and their application as an adsorbent in solid phase extraction. Only one element was chosen as an analyte for extraction. Determination of Cu was performed by AAS. Whereas preconcentration and extraction of heavy metals from water is an important task, the aim of this work is unclear. For the extraction the authors took high concentrations, which can be easily determined by AAS, ICP-OES or ICP-MS directly from water samples. Therefore, it is unclear why the extraction step is needed for analysis. Extraction of trace levels or high adsorption capacity are not shown.
Lines 16-17. Was the sensitivity increased? I cannot see this.
Line 32. Direct analysis of water samples is possible in many cases.
Section 2.2.2. Please indicate experimental parameters for XRD.
Lines 156-157. Did the authors calculate these lattice parameters? The quality of XRD pattern is rather low.
Figure 1. Please add standard XRD data as vertical lines. There is one not identified peak in the upper XRD pattern.
Lines 171-172. What are the symbols used for temperature?
Table 1. The authors state that pore diameter is 9 nm, at the same time they claim that inner diameter of the tubes was 4-6 nm. Is it reliable?
Figure 2b. There is obviously crystalline area, but the authors claim that the walls were amorphous. At the same time the tubes were hollow. What is crystalline part? Which is also seen in XRD.
Table 3. 2.37 or 2.62?
Figure 12. The recovery values were lower with higher mass of the adsorbent; how can it be explained?
Line 327. How the detection limit is related to the extraction procedure? Or it just a characteristic of AAS equipment?
Please calculate the adsorption capacity.
Please use only one type of units – °C or K.
Author Response
Response to Reviewer 4 Comments
The present work by Kefi et al. considers the synthesis of titania nanotubes and their application as an adsorbent in solid phase extraction. Only one element was chosen as an analyte for extraction. Determination of Cu was performed by AAS. Whereas preconcentration and extraction of heavy metals from water is an important task, the aim of this work is unclear.For the extraction the authors took high concentrations, which can be easily determined by AAS, ICP-OES or ICP-MS directly from water samples. Therefore, it is unclear why the extraction step is needed for analysis. Extraction of trace levels or high adsorption capacity are not shown.
Point 1: Lines 16-17. Was the sensitivity increased? I cannot see this.
Response1:
The aim of this work is to test the efficiency of titanium oxide nanotubes as SPE adsorbent of copper from water samples. This application could be applied in the context of water quality control at the laboratory level (identification of copper sources, corrosion control, compliance assessment, estimation of copper exposure, etc.).
On the other hand, the high exposure to copper affects the liver and Symptoms can progress to coma, hepatic necrosis, vascular collapse, and death. The World Health Organization has established a guideline value of 2 mg/l for copper in drinking water (WHO, 2004).
The extraction of copper using titanium oxide nanotubes is carried out under experimental conditions (concentration of copper, pH, pHpzc of the adsorbent, pH of the sample, nature and pH of the extraction solvent, mass of the adsorbent, ...) that favor the adsorption of certain elements such as copper and obviously prevent the adsorption of many other metallic elements and even organic compounds.
Hence we obtain, from complex matrix, an extract where copper is predominant and thus can improve the detection of this element by the analysis technique. On the other hand, these experimental conditions ensure the recovery of copper in small volumes of extraction solvent, which further increases the concentration of this element while remaining it stable and in free form.
Point2: Line 32. Direct analysis of water samples is possible in many cases.
Response 2:
the sentence is corrected in the text as follows: “To analyze them, preliminary treatment of samples containing very complex matrix is thus necessary.”
Point 3: Section 2.2.2. Please indicate experimental parameters for XRD.
Response 3:
More experimental parameters are presented in this section
Point 4: Lines 156-157. Did the authors calculate these lattice parameters? The quality of XRD pattern is rather low.
Response 4:
We confirm values of the lattice parameters, presented in this section. the optimization of the alkaline hydrothermal conditions to elaborate HNTs and TON materials as well as their characterization by XRD are well done in our laboratory. a Reference is added at this result [Hafedh Kochkar, Nesrine Lakhdhar, Gilles Berhault, Marta Bausach, and Abdelhamid Ghorbel, The Journal of Physical Chemistry C 2009 113 (5), 1672-1679. DOI: 10.1021/jp809131z]
Point 5: Figure 1. Please add standard XRD data as vertical lines. There is one not identified peak in the upper XRD pattern.
Response 5:
Standard XRD data as vertical lines are added and peak in the upper XRD pattern is identified.
Point 6: Lines 171-172. What are the symbols used for temperature?
Response 6:
Symbol used for temperature is T
Point 7: Table 1. The authors state that pore diameter is 9 nm, at the same time they claim that inner diameter of the tubes was 4-6 nm. Is it reliable?
Response 7:
HNT used as adsorbent for Cu(II) presented an average pore diameter of 9 nm which is determined by the BET method and outer diameters between 6 and 8 nm and inner diameters of 4–6 n that were determined by TEM analyses. HNT are multiwalled and inner diameter depends on the number of layers.
This characterization is repeated with other elaborated HNT. Nanotubes consisted of several parallel layers spaced from 0.25 to 0.37 nm. The average diameter of HNT is 10 nm with approximate inner diameter of 7 nm.
Point 8: Figure 2b. There is obviously crystalline area, but the authors claim that the walls were amorphous. At the same time the tubes were hollow. What is crystalline part? Which is also seen in XRD.
Response 8:
HNT contains about 18 % H2O more likely located in the interlayer spacing [Hafedh Kochkar, Nesrine Lakhdhar, Gilles Berhault, Marta Bausach, and Abdelhamid Ghorbel, The Journal of Physical Chemistry C 2009 113 (5), 1672-1679. DOI: 10.1021/jp809131z]. The reduction of the spacing and smaller interlayer distance could be attributed to dehydration of the material during TEM observation. Thus The HR-TEM image of HNT shows the coexistence of amorphous and crystalline nanotubes.
Point 9: Table 3. 2.37 or 2.62?
Response 9:
2.37
Point 10: Figure 12. The recovery values were lower with higher mass of the adsorbent; how can it be explained?
Response 10:
A slight decrease is observed but it is insignificant.
Sentence in line 307 is rectified: “The results given in Figure 12 show insignificant differences between calculated yields.”
Point 11: Line 327. How the detection limit is related to the extraction procedure? Or it just a characteristic of AAS equipment?
Response 11:
The detection limit is related to the AAS technique.
Point 12: Please calculate the adsorption capacity.
Response 12:
Adsorption capacity is added in sections 3.3
Point 13: Please use only one type of units – c or K.
Response 13:
°C

Round 2
Reviewer 3 Report
My comments have been well addressed.
Reviewer 4 Report
accept